# Institutional Pressures and Environmental Management Accounting Adoption: Do Environmental Strategy Matter?

**Musaab Alnaim** [1] **and Abdelmoneim Bahyeldin Mohamed Metwally** [1,2,*] 

[1] Department of Accounting, College of Business Administration, King Faisal University, Al-Ahsa 31982, Saudi Arabia; malnaim@kfu.edu.sa
[2] Department of Accounting, Faculty of Commerce, Assiut University, Assiut 71515, Egypt
*   Correspondence: abmetwally@kfu.edu.sa

**Abstract:** This paper examines the impact of institutional pressures (IPs) on Environmental Management Accounting adoption (EMA). The current research also aims to examine the moderating effect of environmental strategy (ES) on the relationship between IP and EMA. Data were collected from managers working in all registered Egyptian manufacturing companies (N = 491). The collected data were analyzed using smart partial least squares (Smart-PLS) software. The results revealed that there is a positive significant relationship between IP's three components, namely, coercive, normative, and mimicry pressures, and EMA. The results also revealed that ES was found to moderate the relationship between IP and EMA. The study model was able to explain 68.9% of the variance in EMA adoption. The findings of this study serve as a pivotal yardstick for guiding corporate policy formulation, offering valuable insights to drive continuous improvements in EMA, environmental performance, and sustainable development. The present investigation extends the discourse on the role of IP and ES by revealing a substantial influence on EMA adoption. Positioned as one of the initial studies to delve into the moderating role of ES in the relationship between IP and EMA adoption, this research offers insights within an emerging market context.

**Keywords:** environmental management accounting; institutional pressures; environmental strategy; sustainability; emerging economy; Egypt

## 1. Introduction

The world is witnessing a surge in environmental challenges, fueled by excessive resource consumption and intensified industrial operations [1,2]. Organizations are pressured by stakeholders to manage harmful waste and proactively participate in environmental preservation efforts, recognizing that maintaining a competitive edge hinges on environmental responsibility [3]. The expansion of industrial activities has played a significant role in exacerbating issues like air and water pollution, as well as rising temperatures [4]. Manufacturing firms are under mounting pressure to prioritize environmental considerations to avoid losing stakeholder support and market share [5,6]. Failure to adapt to eco-friendly practices not only risks alienating customers and investors but also threatens the sustainability of the business [5,7,8].

Despite the potential financial challenges, research strongly advocates for prioritizing sustainable resource management and building organizational resilience [9,10]. This prompts manufacturing companies to actively participate in Corporate Social Responsibility (CSR) activities and innovate in product development to meet the growing demand for eco-friendly initiatives from stakeholders [11]. Such pressures push organizations to place a premium on green innovations, aiming to bolster their environmental performance and secure competitive edges in the market [6].

In recent years, there has been a surge in scholarly interest in CSR and environmental performance. Researchers are focused on assessing the effectiveness of transparency in

disclosing both financial and non-financial aspects of environmental commitments [12]. Previous studies, particularly in less developed countries (LDCs), have linked CSR practices and environmental performance to legitimizing corporate actions [13], influencing management decisions [14], improving firm financial performance [15], fostering employee creativity [16], addressing social and political issues, which result in varied practices [13], and potentially reducing costs, as some argue that CSR disclosure might serve as a rhetorical strategy to mask cost reduction efforts [12].

As a result of this trend, stakeholders are increasingly urging management to prioritize environmental concerns and assess environmental and financial performance [17,18]. To achieve this goal, many companies are considering the adoption of sustainability and its related environmental strategies [8,19]. One prominent approach is the implementation of EMA to enhance their environmental performance [20,21]. However, in most LDCs, especially Egypt, environmental performance reporting remains voluntary, leading to a lack of transparency with limited disclosures [22,23], highlighting the need for urgent further investigation [5,24].

EMA focuses on providing both monetary and physical environmental information to optimize natural resource efficiency and mitigate environmental impact [25,26]. By revealing environmental costs often overlooked in traditional management accounting, EMA supports evidence-based decision-making for top management on environment-related issues [27,28]. EMA influences costing strategies, pricing mechanisms, and production decisions (including those focused on reducing hazardous waste generation) [26,29]. Ultimately, successful EMA adoption and implementation are linked to improvements in both environmental and overall firm performance. Yet, EMA implementation levels remain low with few organizations fully utilizing it for strategic decision-making [30]. This underscores a critical need to investigate the drivers of EMA adoption, especially in developing countries where research focus remains comparatively low [31–33]. Moreover, most existing studies draw on legitimacy, agency, and stakeholder theories to explain EMA adoption, with limited use of institutional theory [34]. This paper seeks to address these gaps by applying institutional theory to examine drivers of EMA adoption in Egypt. Specifically, the empirical focus is on whether coercive, mimetic, and normative pressures significantly influence EMA adoption decisions [33].

Institutional theory, especially new institutional sociology, offers a valuable lens through which researchers are able to examine the drivers motivating organizations to adopt EMA [5,7,35]. As institutional theory posits that organizational behaviors and practices are not solely determined by internal factors but are significantly influenced by the institutional environment [5,7]. This environment encompasses a complex web of formal and informal elements, including laws, values, cultural norms, and societal expectations [36]. Coercive pressure (the first driver) suggests that powerful actors (governments, suppliers, etc.) enforce compliance via coercive isomorphism, shaping organizational behavior [35,36]. Normative pressure (the second driver) originates from shared organizational norms, pushing firms to conform [35,36]. Finally, mimetic pressure prompts firms to imitate successful peers in uncertain environments [35,36].

Research examining the moderating role of environmental strategies (ESs) on the relationship between IP and EMA adoption remains sparse. Furthermore, the combined effects of IP and ES in driving EMA adoption are under-explored, with the notable exception of ref. [5] who found that IP and ES independently impact EMA adoption positively. This study aims to fill this empirical gap and contribute to the existing body of knowledge. A unique aspect of this research is the extension of the investigation to the understudied Egyptian industrial sector. Additionally, it examines the moderating role of ES in the IP–EMA adoption relationship, offering a novel contribution as it potentially confirms or challenges prior findings within a different African emerging market context. This study seeks to answer the following research questions: (1) How do IPs impact the adoption of EMA? (2) Does the presence of ES moderate the relationship between IP and EMA adoption?

This study's proposed research framework offers a novel contribution to the literature by investigating the factors that influence the adoption of EMA. It further extends the understanding by examining how ES potentially moderates the relationship between IP and EMA adoption. The significance of this research is magnified by its focus on validating the framework within a key industrial sector in the developing context of Egypt. The rest of this paper is structured as follows: Section 2 outlines the theoretical framework employed in the study. Section 3 analyzes existing literature to identify research gaps and develops study hypotheses. Section 4 details the research methodology and methods adopted. Section 5 presents the findings and offers a critical discussion. Section 6 presents a comprehensive discussion and conclusion; and the final section addresses implications, limitations, and avenues for future research.

## 2. Theoretical Framework

EMA adoption and its impact on environmental performance are studied heavily in the management accounting literature through deploying many theoretical frameworks including the following: agency theory, stakeholders theory, and institutional theory [33,37]. From a theoretical standpoint, the stakeholder theory posits that environmental strategies are primarily adopted for value creation, aligning with shareholder interests and ultimately enhancing their wealth [38]. Empirical evidence supports this view, suggesting a positive association between environmental initiatives and organizational benefits [37,39]. Conversely, the agency theory proposes that managers may engage in environmental discourse and initiatives for personal gain, potentially leading to conflicts of interest with stakeholders [12,40]. This perspective suggests that managers might prioritize reputation management over actual environmental impact, potentially engaging in CSR washing [41]. Additionally, research suggests that sustainability practices could be linked to reduced capital allocation efficiency [42], potentially incurring agency costs. Further, CSR initiatives might be strategically employed to mitigate potential repercussions of management decisions [37,43].

Moreover, stakeholder theory argues that companies must actively engage with their stakeholders to achieve sustainable performance and gain a competitive edge in the market [44]. This theory emphasizes the importance of building strong relationships with stakeholders, which can potentially lead to various benefits such as cost savings, reduced environmental impact, and improved overall performance [33,44]. Additionally, stakeholder engagement is proposed to mitigate environmental uncertainties, yielding benefits like better product and service management, attracting and retaining high-quality employees, enhancing company reputation and customer loyalty, and ultimately ensuring sustained competitive advantages [45–48]. Notably, the implementation of EMA practices is seen as a tool for reducing environmental uncertainties [49]. This reduction is believed to subsequently improve the utilization of both tangible and intangible assets, ultimately contributing to both environmental and economic performance within organizations [33,49].

Other studies in the literature concentrate on understanding EMA adoption and impact from an institutional theory perspective. However, many divergent views are discussed in this regard as some researchers have followed different versions of institutional theory including old institutional economics; new institutional economics; new institutional sociology and institutional logics [50–52]. Institutional theory, especially new institutional sociology, offers a valuable lens through which researchers are able to examine the drivers motivating organizations to adopt EMA [5,7,35].

This theoretical framework posits that organizations, in their pursuit of both survival and legitimacy, often exhibit a tendency to conform to established and prevailing practices, regardless of their immediate effectiveness within the specific organizational context [53]. Such conformity is viewed as a means to achieve increased stability, legitimacy, and, ultimately, enhanced access to resources [54,55]. Further, it explains that companies are integrated into a system of shared values, standards, conventions, and beliefs that determine acceptable behavior. These societal structures, or institutions, become ingrained over time

and provide validated models for action [22,56]. To increase their chances of survival, managers often adapt to these institutions, aligning their practices with social expectations to gain legitimacy [36,57,58].

Seeking legitimization has both strategic and institutional dimensions [7]. Strategic theorists view legitimacy as a manageable resource, with managers exercising control over how it is achieved. In contrast, institutional theorists see legitimacy as a set of beliefs that shape how an organization is founded, operated, and perceived [59]. To gain legitimacy, organizations may need to transform their structure, culture, goals, or mission [36]. This pressure towards conformity can lead to isomorphism, a process where similar organizations within the same environment adopt identical traits [7,36].

New institutional sociology (NIS) has been used to explain EMA adoption patterns [33,50,60]. NIS argues that organizations seek legitimacy by adapting management practices, aligning with institutionalized norms and expectations [33,55,60]. To be seen as legitimate, they may adopt practices promoted by powerful stakeholders, becoming part of organizational rituals [59]. Isomorphism can be competitive or institutional [7]. In a free and competitive market, organizations face competitive isomorphism, a pressure to adopt similar efficiency-enhancing practices due to competition for resources. This leads to a degree of homogenization among organizations [7]. Institutional isomorphism, on the other hand, arises from external pressures that force changes without necessarily aiming for efficiency. This can occur through three mechanisms: coercive pressure (COP) from governing bodies, mimetic pressure (MIP), imitation of successful competitors, or normative pressure (NOP) from professional bodies [7,36].

## 3. Literature Review and Hypotheses Development

### 3.1. Institutional Pressures and EMA Adoption

Institutional theory explains how organizations, including their energy use, environmental practices, and management strategies, are shaped by their environment [57]. Companies are influenced by external factors like laws, regulations, societal values, and cultural expectations [36,61]. To remain sustainable, they must adapt to these changes, as ignoring them could be detrimental [36,62]. Therefore, acknowledging external shifts and implementing EMA principles become crucial [63].

Institutional pressures, which influence organizational behavior, come in three forms. COP is exerted by powerful stakeholders like governments, NGOs, customers, and suppliers, often involving strict regulations and penalties [36,61]. NOP stems from internal values, company culture, and professional standards. This pressure pushes organizations towards adopting new practices [62]. Finally, MIP results from uncertainty. Companies often mimic successful competitors or react to internal and external changes [33,64].

In the domain of institutional theory, the integration of EMA within corporate frameworks is subject to the exertion of COP, NOP, and MIP [5,7,35,55,60,65]. Despite the initial presumption of these pressures facilitating organizational change, scholarly investigations underscore the prevalent occurrence of decoupling as a strategic response to IP [13,22,51,56,66,67]. Decoupling manifests when these pressures fail to catalyze or expedite transformative processes, particularly in contexts where internal social norms or the lack of robust exemplars attenuate the impact of COP [22,51]. Regrettably, such circumstances often engender nominal effects on change endeavors or even adversarial ramifications on overall organizational performance, as entities resort to covert resistance and superficial adherence to perpetuate the existing operational paradigms [13,22,51,56,66,67].

### 3.1.1. Coercive Pressure and EMA Adoption

Institutional theory highlights the role of COP, exerted by external stakeholders like governments and NGOs, in shaping environmental regulations and standards for companies. These regulations are mandatory and directly influence organizations' environmental protection efforts and legislative mandates [68,69]. Although the theory suggests that COP

primarily targets external aspects, its impact can reach internal organizational behaviors as well [33,69].

COP is the most prominent pillar as it is driven by political influence and concerns about legitimacy [35]. This pressure may take formal form (laws and regulations) or/and informal form (societal expectations), exerted by powerful institutions and cultural norms [35,36]. To be perceived as legitimate, organizations may feel forced, persuaded, or implicitly invited to change their behavior and structure [5,35,55,60]. This pressure to conform can lead to the adoption of practices used by other organizations, even without evidence of their effectiveness for the specific organization [7,22].

Developing countries may experience COP to implement EMA from various external actors, such as international buyers in developed countries and foreign investors. This pressure can play a significant role in EMA adoption not only in developing markets but also in developed markets [5,33,55,60,68], yet the scope and impact of this pressure may be different in developed countries from developing countries [13,22,51,56]. Studies have shown that COP, often manifested through mandatory regulations imposed by government authorities, can positively impact companies' environmental performance [36,70]. These regulations are accompanied by potential sanctions for non-compliance [5,7,33,60].

Furthermore, beyond regulatory compliance, EMA adoption can bring additional benefits to companies facing COP. It can help improve their environmental performance, potentially leading to government support and economic advantages [68,69]. Additionally, EMA can enhance a company's social reputation [68,69]. These combined factors contribute to the observed trend of companies adopting EMA practices under COP [5,7,33,35,55,60,71]. Based on this, the following hypothesis is proposed:

**H1:** *A positive association exists between the COP and EMA adoption.*

3.1.2. Normative Pressure and EMA Adoption

NOP, stemming from industry associations, media, and other stakeholders, directly influences EMA adoption. Industry associations, particularly, play a crucial role in establishing norms around EMA implementation. Membership in these associations exposes firms to expectations regarding behavior and can provide access to valuable resources and expertise through networking [35,57,60,64,65]. Firms may choose to conform to these norms to avoid jeopardizing partnerships and access to resources. Additionally, pressure from media and the public can compel firms to implement EMA to address public environmental concerns and avoid reputational damage, which could lead to loss of external resources like bank financing [35,72]. Ultimately, firms may adopt EMA practices to enhance their reputation and secure long-term benefits [5,7,35,65].

Other than industry associations and media, NOPs also arise from suppliers, customers, and social entities like trade unions [35,57]. These actors, particularly trade unions, often act as the primary source of NOP within developing countries, shaping environmental responsibility and ethical behavior [7,33,36,70]. Unlike developed economies, where cooperative relationships among organizations across networks contribute to NOP [36], developing countries experience a strong emphasis on individual responsibility and compliance with social norms within organizations [7,33,36,70]. This pressure encourages both organizations and external actors, such as customers and suppliers, to adopt socially responsible practices, including EMA, to maintain good standing and legitimacy.

Further, companies implement EMA as one of the strategies that minimize negative impacts on trade unions, recognizing their influence on internal resources, knowledge, and company culture [33]. Through EMA adoption and effective communication practices, companies can manage public perception and build a positive reputation [33,35,60]. Conversely, neglecting public perception or resisting change through unions can damage the company's image, leading to potential losses in external resources and competitive disadvantage [5,7,33,55,60]. Therefore, EMA adoption influences a company's reputation,

performance, competitive advantage, and ultimately, its overall image [33]. Based on this, the following hypothesis is proposed:

**H2:** *A positive association exists betweenthe NOP and EMA adoption.*

### 3.1.3. Mimetic Pressure and EMA Adoption

MIP constitutes the third mechanism driving isomorphism within the framework of institutional theory [73]. This concept of mimetic isomorphism suggests that, to cope with environmental uncertainties, organizations often imitate perceived success by adopting practices employed by established and successful counterparts, thereby seeking legitimacy and promoting institutional isomorphism [36,74]. This convergence of practices is seen as a path to achieve legitimacy within the organizational field [75,76].

Mimetic isomorphism is a process demonstrably influenced by cultural norms [75], which entails that organizations facing environmental uncertainties tend to emulate the practices of successful competitors [77]. This phenomenon manifests in three primary forms, namely, frequency-based imitation, which is the basic form, where organizations simply replicate practices widely adopted by the majority within their industry [78]; trait-based imitation, which is a more selective approach, where organizations specifically focus on imitating practices observed in other firms exhibiting desired characteristics, such as size or prominence within the field [78,79]; and finally, outcome-based imitation which is similar to trait-based imitation; this form involves selective imitation of practices associated with positive outcomes in other firms [75,78].

These various forms of MIP highlight the diverse pathways through which organizations can learn and adapt by replicating prevalent practices within their environments [75]. Ultimately, this imitation contributes to the phenomenon of institutional isomorphism, where organizations within a field exhibit increasing homogeneity in their practices. As mentioned earlier, the specific practices and the degree to which they are imitated are likely influenced by cultural norms and expectations [60,75].

In the context of EMA adoption, where implementation can be costly and financial returns uncertain [33,70], learning from successful competitors becomes particularly valuable. MIP can thus drive firms to imitate the practices of successful EMA adopters, motivated by the potential to reduce exploration costs associated with researching alternative approaches and minimize experimentation costs associated with implementing untested solutions [7,33,35,55,60,70].

The influence of MIP on EMA adoption exhibits significant variation across different contexts [33]. In developing countries, MIP often encourages foreign-owned and multinational corporations to implement more robust environmental management practices [33,60]. These practices can then serve as a model for local organizations, potentially leading to improved environmental performance within the region [36].

In contrast, developed economies like Europe and North America view MIP as a strategic tool for achieving superior performance [33,70]. Companies in these regions may adopt or utilize green technologies and resources in response to international demands and stakeholder pressures, aiming to gain a competitive edge [7,33,35].

Furthermore, strong MIP can influence governments and stakeholders to encourage the adoption of advanced environmental management practices and technologies by local companies [33]. This can be achieved by promoting the benefits observed in foreign-owned or MNC subsidiaries, potentially leading to improved performance across the entire region [36,57]. Ultimately, companies that respond to MIP by adopting EMA, despite the associated costs, can gain economic benefits through increased competitiveness [33]. Therefore, EMA adoption serves as a strategic response to MIP, potentially leading to both environmental and economic advantages [5,33,35,55,60,65]. Based on this, the following hypothesis is proposed:

**H3:** *A positive association exists between mimicry pressures and EMA adoption.*

### 3.2. The Impact of Environmental Strategy: ES Moderating Role

Corporate environmental strategies (ESs) encompass a series of organizational initiatives aimed at minimizing environmental impact across product life cycles, operational processes, and corporate policies [19,80–82]. These strategies represent a conscious effort to integrate business activities with environmental considerations, fostering a more sustainable approach [5,83]. The implementation of comprehensive environmental management systems serves as a key driver for translating environmental strategies into tangible actions. These systems facilitate the development of corporate environmental proactivity, defined as the "voluntary implementation of practices and initiatives aimed at improving environmental performance" [5,84].

Organizations exhibit varying levels of commitment and proactivity when implementing ES. This variance has led researchers to conceptualize a continuum that ranges from a reactive (passive) stance to a proactive environmental leadership position [5,83,85]. Mårtensson and Westerberg [86] emphasize that organizational factors such as material flow, knowledge transfer, relationships, communication dynamics, cooperation, and control mechanisms influence the way a firm adopts and executes an ES. Specifically, accounting information systems like those focused on EMA play a vital role in driving ES implementation [5]. EMA provides critical data on environmental costs and enables the integrated monitoring of environmental and financial performance metrics [5,70,85,87].

Prior empirical research has explored the link between ES and EMA through the lens of contingency theory [71,88,89]. However, a comprehensive investigation into the isolated impact of ES on EMA adoption (as a contextual factor) remains a gap in the literature, particularly when the potential effects on corporate performance are excluded. Results from existing studies offer conflicting evidence on the relationship between ES and EMA. Some studies indicate a significant association between ES and EMA practice implementation [5,24,71,88,90], while others suggest no meaningful relationship [5,91,92].

While numerous studies identify a positive correlation between pursuing a "green" strategy and organizational performance, others caution that such benefits only materialize under specific conditions [81,87,93]. For instance, organizations that cultivate green capabilities, such as robust sustainability control systems exemplified by EMA, are better positioned to achieve a sustainable competitive advantage. It is with this aim—realizing these environmental and financial benefits—that organizations implementing environmental management practices often seek to develop environmental accounting and control systems [70]. This aligns with the established principle that different organizational strategies necessitate specific accounting information systems to cater to the unique needs of decision-makers [5]. This rationale is further supported by research demonstrating a significant link between organizational strategies and the adoption of specific EMA [5,24,71,88,90].

Having said this, there is a notable scarcity in the examination of the moderating influence of ES on the interplay between IP and EMA adoption. Nevertheless, antecedent research has underscored and established a direct relationship between ES and EMA adoption. Aligning with theoretical discourse and precedent empirical evidence, it is anticipated that ES will exert a discernible influence on the relationship between IP and EMA adoption within the Egyptian market. Accordingly, we posit the following hypotheses:

**H4:** *ES moderates the relationship between COP and EMA adoption.*

**H5:** *ES moderates the relationship between NOP and EMA adoption.*

**H6:** *ES moderates the relationship between MIP and EMA adoption.*

The relationship between the study variables is presented in Figure 1 below.

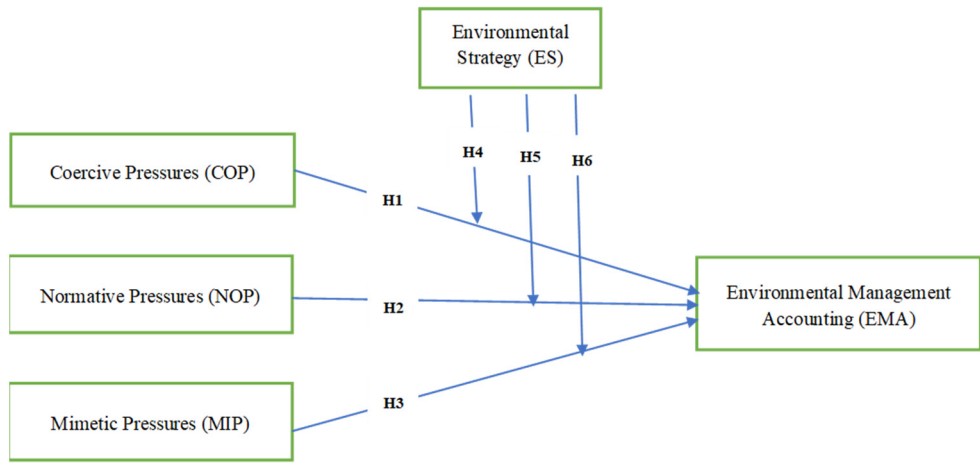

**Figure 1.** Study framework model.

## 4. Methodology

### 4.1. Data Collection and Survey Design

The research focused on manufacturing firms listed on the Egyptian Stock Exchange (ESE) as the sample for the study. The choice of the manufacturing sector stemmed from its significant environmental impact and the prevalent environmental challenges faced by these companies. The study anticipated that various IPs exist, compelling manufacturing companies in Egypt to adopt environmental practices. This underscores the relevance of EMA adoption for manufacturing companies within the Egyptian context. The list of manufacturing firms was sourced from the ESE. In all, 650 questionnaires were given out to managers in these manufacturing companies, given their expertise in both accounting and a strategic perspective on the implementation of EMA and ES, as noted by [5,33,92]. The survey was conducted through web-based and manual methods. The survey was presented to a group of accounting lecturers who are knowledgeable with this area as well as three professional accountants. Drawing from their insights, we implemented various modifications to enhance the questionnaire's clarity and comprehensibility, adjusting instructions, question order, and wording accordingly. The survey utilized a 5-point Likert scale, where respondents could express their agreement or disagreement on a spectrum from 1 (completely disagree) to 5 (completely agree). This scale was employed to gauge participants' responses to the questionnaire items. This study initiated data collection in October 2023, spanning a four-month period. Among the distributed questionnaires, 510 were returned. After excluding 19 incomplete submissions, the final dataset for analysis consisted of 491 questionnaires, yielding a response rate of 75.5% (491 out of 650). Demographic analysis of the respondents is presented in Table 1.

**Table 1.** Respondents' profiles.

|  |  | **Freq.** | **%** |
|---|---|---|---|
| Gender | Male | 355 | 72.3 |
|  | Female | 136 | 27.7 |
| Experience (years) | 1–5 years | 16 | 3.2 |
|  | 6–10 years | 131 | 26.7 |
|  | 11–15 years | 151 | 30.7 |
|  | More than 15 years | 193 | 39.4 |
| Educational Level | Bachelor's degree | 192 | 39.1 |
|  | Post-graduate Degree | 299 | 60.9 |

**Table 1.** *Cont.*

|  |  | Freq. | % |
|---|---|---|---|
| Industry Type | Industrial Goods, Services, and Automobiles | 84 | 17 |
| | Basic Resources | 73 | 14.9 |
| | Healthcare and Pharmaceuticals | 60 | 12.3 |
| | Food, Beverages, and Tobacco | 135 | 27.5 |
| | Building Materials | 103 | 21 |
| | Textile and Durables | 36 | 7.3 |
| | Total | 491 | 100 |

*4.2. Measures and Scale Development*

The survey utilized in this study was structured into three distinct sections, each meticulously tailored to meet the specific demands of the research. Initially crafted based on insights from prior studies, the instrument underwent refinements to align seamlessly with the contextual nuances of the present investigation. The initial section centered on obtaining explicit informed consent from study participants, wherein individuals explicitly conveyed their voluntary agreement to partake in the research. Then, it was followed by collecting demographic information about the respondents. Subsequently, the second segment featured inquiries pertaining to institutional pressures, categorized into coercive, normative, and mimetic, drawing from the research works of [33,35,57,69,94]. The third part of the instrument centered on queries related to EMA and ES. Six questions, adapted from studies conducted by [33,71,95,96], were employed for the measurement of EMA. To assess the ES scale, an instrument provided by [5,24] was employed. Table 2 succinctly outlines the principal constructs and their operationalization methods.

**Table 2.** Measurement model.

| Scale Variables and Items | Outer Loading | Alpha | CR | AVE |
|---|---|---|---|---|
| **Coercive Pressures (COP)** | | 0.860 | 0.873 | 0.702 |
| Our company endeavors to mitigate the threat from environmental regulations by incorporating environmental management accounting. | 0.842 | | | |
| Our company considers environmental regulations to be crucial in driving the implementation of environmental management accounting. | 0.804 | | | |
| Our company must adhere to the stringent environmental regulations established by the local government. | 0.891 | | | |
| Companies that break environmental standard and regulations face a number of penalties. | 0.812 | | | |
| **Normative Pressures (NOP)** | | 0.859 | 0.862 | 0.703 |
| Our company has been prompted to adopt environmental management accounting due to the growing environmental awareness among consumers. | 0.794 | | | |
| For our company to be part of this industry, it is fundamentally necessary to provide environmental information and being environmentally responsible. | 0.879 | | | |
| The nongovernmental organizations in our community expect that environmental management accounting be used by all companies in the industry. | 0.826 | | | |
| Without implementing environmental management accounting, stakeholders might not support our company. | 0.853 | | | |

**Table 2.** *Cont.*

| Scale Variables and Items | Outer Loading | Alpha | CR | AVE |
|---|---|---|---|---|
| **Mimetic Pressures (MIP)** | | 0.883 | 0.886 | 0.740 |
| Leading firms in our industry serve as role models for the application of environmental management accounting. | 0.840 | | | |
| It is common knowledge that the top firms in our industry successfully implementation of environmental management accounting. | 0.865 | | | |
| The top firms in our industry intent to use environmental management accounting to reduce their environmental effects. | 0.873 | | | |
| Implementing environmental management accounting has given the top firms in our industry a competitive edge. | 0.862 | | | |
| **Environmental Strategy (ES)** | | 0.946 | 0.948 | 0.787 |
| Our company's environmental strategic plan promotes sustainable resource management and creates a long-term commitment to the environment. | 0.876 | | | |
| Our company's environmental strategic plan works to reduce the environmental impacts of products and services. | 0.892 | | | |
| Our company's environmental strategic plan employs environmental management systems. | 0.915 | | | |
| Our company's environmental strategic plan sets performance indicators to measure the level of pollution and reduce emissions (air, water, energy, waste). | 0.858 | | | |
| Our company's environmental strategic plan seeks to invest in research and development activities related to environmental protection. | 0.922 | | | |
| Our company's environmental strategic plan is working towards obtaining ISO certificates and environmental awards. | 0.858 | | | |
| **Environmental Management Accounting (EMA)** | | 0.943 | 0.945 | 0.779 |
| The accounting system of our firm diligently captures and records all physical inputs and outputs, encompassing such as energy, water, materials, wastes, and emissions. | 0.851 | | | |
| The accounting system utilized by our firm is capable of conducting product inventory analyses, product improvement analyses, and assessments of product environmental impacts. | 0.929 | | | |
| Our firm employs environmental performance targets for monitoring and managing physical inputs and outputs. | 0.918 | | | |
| Environmentally linked costs and liabilities can be recognized, estimated, and categorized by our company's accounting system. | 0.887 | | | |
| The accounting system within our firm has the capability to establish and utilize Cost accounts relating to the environment. | 0.868 | | | |
| The accounting system employed by our firm has the capability to allocate environmental-related costs to products. | 0.840 | | | |
| Average variance extracted (AVE) and composite reliability (CR). | | | | |

### 4.3. Data Analysis Methods

In our analytical endeavors, we employed the Partial Least Squares Structural Equation Modeling (PLS-SEM) technique using SmartPLS-4. Our initial procedures encompassed assessments aimed at gauging the reliability and validity of the utilized instruments. Subsequently, for the examination of proposed hypotheses, Structural Equation Modeling (SEM) analyses were conducted. Within the realm of management research, PLS path modeling stands out as a robust method for computing intricate cause-and-effect connection models, as underscored by [97]. PLS-SEM analyses confer the advantage of facilitating

solutions for highly complex models that involve numerous constructs, indicators, and structural relationships [98]. This methodology is particularly well-suited for the initial development and testing of theories [99], enabling the exploration of relationships and constructs within intricate structural models.

Significantly, PLS-SEM exhibits efficacy with smaller sample sizes, operates efficiently in handling complex models, and avoids assumptions about data distribution [98]. When employing PLS-SEM, meticulous attention must be given to ensuring the validity and reliability of the measurement model (outer model), followed by the testing of relationships and hypotheses through the structural model (inner model).

## 5. Results

### 5.1. Measurement Model Assessment

The intent behind assessing the measurement model was to ascertain the validity and reliability of the constructs, as delineated in Table 2. Every item within the model demonstrated factor loadings surpassing the established minimum threshold of 0.708 and exhibited statistical significance. Additionally, Table 2 presents alpha Cronbach ($\alpha$), composite reliability (CR), and average variance extracted (AVE) values, all surpassing the respective threshold values ($\alpha$ and CR > 0.7, AVE > 0.5), confirming convergent validity. Notably, the Variance Inflation Factor (VIF) values for the measurement scale items were consistently below 5, indicating the absence of concerns regarding multicollinearity [99]. These results collectively affirm the reliability and validity of the employed measures. Outer-loadings for every latent variable greatly exceeded cross-loadings, as seen by Table 3's results. Table 4 illustrates the process of establishing discriminant validity by comparing correlations among latent variables with the square root of AVE [100] and the Heterotrait–monotrait ratio (HTMT) of correlations [97], all of which fell below the conservative threshold of 0.90. Consequently, discriminant validity was validated.

**Table 3.** Cross-loadings indicators.

|       | COP   | EMA   | ES    | MIP   | NOP   |
|-------|-------|-------|-------|-------|-------|
| COP-1 | 0.842 | 0.613 | 0.422 | 0.459 | 0.534 |
| COP-2 | 0.804 | 0.498 | 0.416 | 0.557 | 0.398 |
| COP-3 | 0.891 | 0.600 | 0.406 | 0.542 | 0.589 |
| COP-4 | 0.812 | 0.646 | 0.426 | 0.548 | 0.634 |
| EMA-1 | 0.601 | 0.851 | 0.420 | 0.592 | 0.571 |
| EMA-2 | 0.633 | 0.929 | 0.464 | 0.596 | 0.631 |
| EMA-3 | 0.595 | 0.918 | 0.418 | 0.582 | 0.608 |
| EMA-4 | 0.554 | 0.887 | 0.446 | 0.507 | 0.571 |
| EMA-5 | 0.569 | 0.868 | 0.398 | 0.545 | 0.569 |
| EMA-6 | 0.565 | 0.840 | 0.430 | 0.502 | 0.514 |
| ES-1  | 0.437 | 0.469 | 0.876 | 0.386 | 0.393 |
| ES-2  | 0.427 | 0.464 | 0.892 | 0.401 | 0.405 |
| ES-3  | 0.399 | 0.412 | 0.915 | 0.465 | 0.358 |
| ES-4  | 0.457 | 0.420 | 0.858 | 0.395 | 0.359 |
| ES-5  | 0.475 | 0.422 | 0.922 | 0.502 | 0.392 |
| ES-6  | 0.454 | 0.391 | 0.858 | 0.422 | 0.316 |
| MIP-1 | 0.639 | 0.622 | 0.359 | 0.840 | 0.487 |
| MIP-2 | 0.519 | 0.574 | 0.392 | 0.865 | 0.426 |
| MIP-3 | 0.597 | 0.624 | 0.462 | 0.873 | 0.550 |

**Table 3.** *Cont.*

|       | COP   | EMA   | ES    | MIP   | NOP   |
|-------|-------|-------|-------|-------|-------|
| MIP-4 | 0.542 | 0.613 | 0.436 | 0.862 | 0.581 |
| NOP-1 | 0.437 | 0.535 | 0.301 | 0.451 | 0.794 |
| NOP-2 | 0.603 | 0.591 | 0.389 | 0.556 | 0.879 |
| NOP-3 | 0.594 | 0.518 | 0.324 | 0.460 | 0.826 |
| NOP-4 | 0.568 | 0.557 | 0.388 | 0.534 | 0.853 |

**Table 4.** Discriminant validity measures of scales.

|          | Fornell-Larcker | | | | | HTMT | | | | |
|----------|-------|-------|-------|-------|-------|-------|-------|-------|-------|-------|
|          | COP   | ES    | EMA   | MIP   | NOP   | COP   | ES    | EMA   | MIP   | NOP   |
| 1. COP   | 0.838 |       |       |       |       |       |       |       |       |       |
| 2. ES    | 0.498 | 0.887 |       |       |       | 0.553 |       |       |       |       |
| 3. EMA   | 0.778 | 0.487 | 0.883 |       |       | 0.845 | 0.513 |       |       |       |
| 4. MIP   | 0.785 | 0.481 | 0.742 | 0.860 |       | 0.831 | 0.526 | 0.809 |       |       |
| 5. NOP   | 0.657 | 0.419 | 0.657 | 0.559 | 0.839 | 0.747 | 0.462 | 0.728 | 0.680 |       |

### 5.2. Hypotheses Testing

This research comprises six hypotheses, categorized into three direct-influencing hypotheses (H1, H2, and H3) and three hypotheses (H4, H5, and H6) designed to assess the moderating impact of ES. Figure 2 provides a summary of the results from the inner model test, displaying estimations of paths and causal connections linking IP (COP, NOP, and MIP) and EMA, incorporating ES as a moderating factor. Hypothesis validation relies on the significance of path coefficients (β); acceptance occurs when these coefficients' values are statistically significant. In PLS-SEM, a hypothesis is deemed accepted if the t-value exceeds 1.96, equivalent to $p < 0.05$.

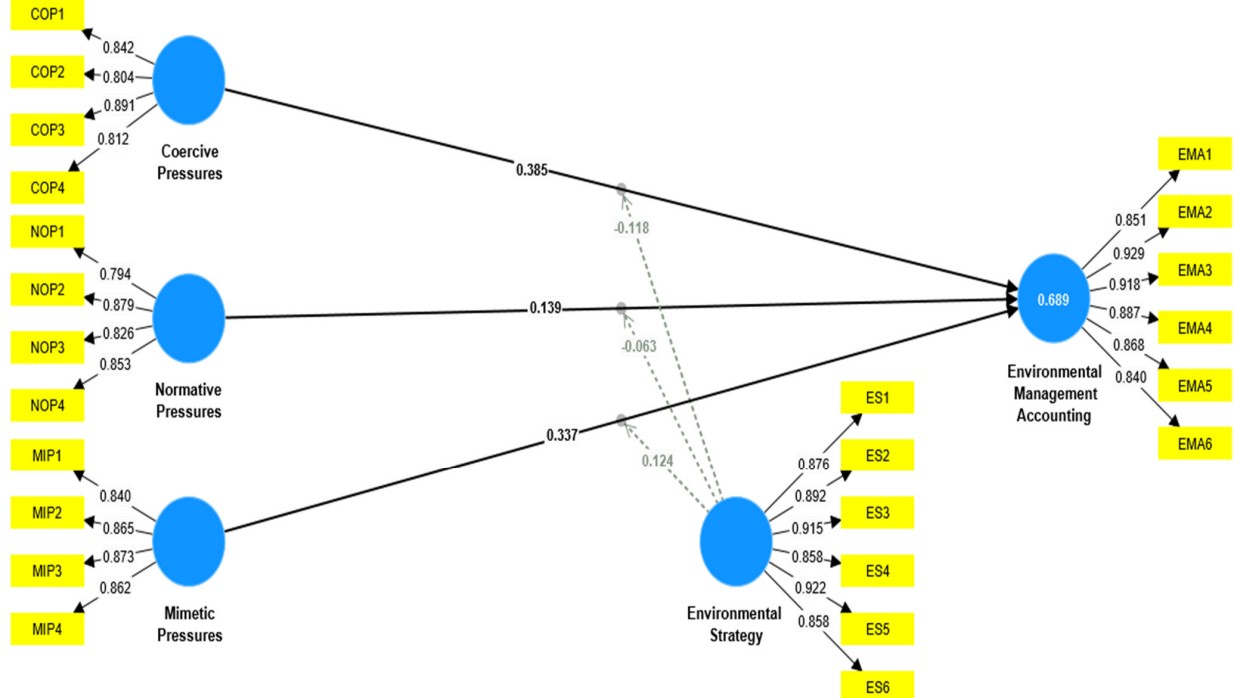

**Figure 2.** Final research model.

Table 5 displays the results of the hypothesis testing. The findings provided support for this study's proposed hypotheses (H1, H2, and H3), which pertain to the direct influence of IP on EMA. Specifically, for the initial hypothesis (H1), COP exhibited a positive association with EMA ($\beta$ = 0.385; t-value = 6.366), establishing its statistical significance and acceptance. In a parallel manner, the findings pertaining to the second hypothesis (H2) indicated a positive association between NOP and EMA ($\beta$ = 0.139; t-value = 3.386), attaining statistical significance and thereby warranting acceptance. Similarly, the third hypothesis (H3) uncovered a positive relationship between MIP and EMA ($\beta$ = 0.337; t-value = 5.049), establishing its statistical significance and meriting acceptance.

**Table 5.** Structural parameter estimates.

|  | Hypotheses | Beta ($\beta$) | T-Statistics | Results |
|---|---|---|---|---|
| H-1 | COP → EMA | 0.385 *** | 6.366 | Accepted |
| H-2 | NOP → EMA | 0.139 *** | 3.386 | Accepted |
| H-3 | MIP → EMA | 0.337 *** | 5.049 | Accepted |
|  | ES → EMA | 0.069 * | 2.118 | Accepted |
| H-4 | COP × ES → EMA | −0.118 * | 2.124 | Accepted |
| H-5 | NOP × ES → EMA | −0.063 | 1.015 | Not Accepted |
| H-6 | MIP × ES → EMA | 0.124 * | 2.233 | Accepted |

Note: *** $p < 0.001$, and * $p < 0.05$.

The investigation delved into the scrutiny of hypotheses related to the moderating influence of ES (ES) on the connection between IP and EMA. The outcomes affirmed support for H4 and H6, whereas H5 failed to attain statistical significance, evident in a *p*-value surpassing 0.05. The observed effect of ES on EMA was positive and significantly established ($\beta$ = 0.069; t-value = 2.118). Concerning H4, which explores the moderating impact of the interaction between COP and ES (COP × ES), the outcome manifested as negative and statistically significant ($\beta$ = −0.118; t-value = 2.124), as Figure 3 illustrates that the relationship between COP and EMA was shown to be diminished by ES. Conversely, H5, investigating the moderating impact of the interaction between NOP and ES (NOP × ES), did not achieve statistical significance ($\beta$ = 0.063; t-value = 1.015). Additionally, H6, examining the moderating effect of the interaction between MIP and ES (MIP × ES), showed a positive and statistically significant result ($\beta$ = 0.124; t-value = 2.233), as Figure 4 illustrates that the relationship between MIP and EMA was shown to be increased by ES. The calculated $R^2$ value was 68.9%, indicating a high explanatory percentage and quality of the model, suggesting that the model explains 68.9% of the variance in endogenous constructs.

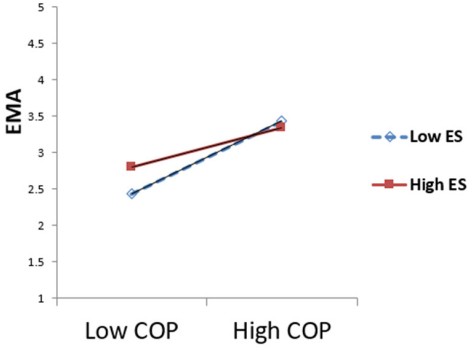

**Figure 3.** ES moderates the relationship between COP and EMA.

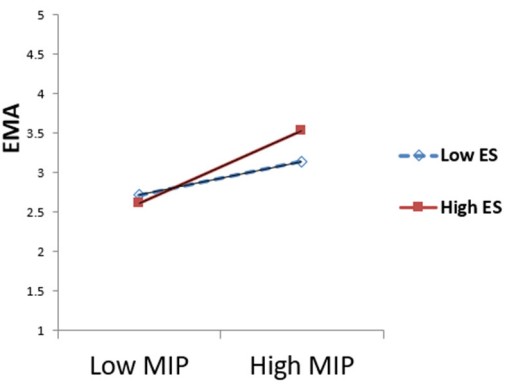

**Figure 4.** ES moderates the relationship between MIP and EMA.

## 6. Discussion and Conclusions

The current study examined the impact of three institutional pressures, namely, COP, NOP, and MIP, on EMA adoption based on the presence of ES, a moderator in the Egyptian industrial sector. The results revealed that there is a positive significant relationship between IP's three components, namely, coercive, normative, and mimicry pressures, and EMA adoption. Hence, the first three hypotheses were accepted. The results also revealed that ES was found to have a negative impact on the relationship between COP and EMA adoption and a positive impact on the relationship between MIP and EMA adoption. The study model was able to explain 68.9% of the variance in EMA adoption.

Moreover, our study confirmed H1, affirming the favorable impact of COP on EMA adoption. Consistent with prior research [5,7,33,35,55,60,71], a positive correlation was evident between COP and EMA adoption. In developing markets like Egypt, to be perceived as legitimate, companies feel forced, persuaded, or implicitly invited to change their behavior and structure [5,35,55,60]. This pressure to conform can lead to the adoption of practices used by other organizations, even without evidence of their effectiveness for the specific organization [7,22]. In that sense, COP is confirmed to be the most powerful driver in EMA adoption in the Egyptian context.

H2, indicating a significant and positive association between NOP and EMA adoption, was supported by our study. The presence of NOP notably augments EMA adoption. Consistent with prior research [5,7,33,35,55,60], our findings align with the established causal link between NOP and enhanced level of EMA adoption. Previous studies have illustrated that firms may choose to conform to surrounding norms to avoid jeopardizing partnerships and access to resources. These norms are established around EMA implementation by industry associations and other stakeholders. Membership in these associations exposes firms to expectations regarding behavior and can provide access to valuable resources and expertise through networking [35,57,60,64,65]. Additionally, pressure from the media and the public can compel firms to implement EMA to address public environmental concerns and avoid reputational damage, which could lead to loss of external resources like bank financing [35,72]. Ultimately, firms may adopt EMA practices to enhance their reputation and secure long-term benefits [5,7,35,65].

Our study further confirmed H3, affirming the favorable impact of MIP on EMA adoption. Consistent with prior research [5,7,33,35,55,60,71], a positive correlation was evident between MIP and EMA adoption. In that sense, firms' tendency to follow and learn from successful competitors becomes particularly valuable, where EMA implementation can be costly and financial returns uncertain [33,70]. Hence, MIP can thus drive firms to imitate the practices of successful EMA adopters, motivated by the potential to reduce exploration costs associated with researching alternative approaches and minimize experimentation costs associated with implementing untested solutions [7,33,35,55,60,70].

The findings of this study regarding the moderating impact of ES are noteworthy, given that H4 and H6 were affirmed while H5 was rejected. As per the experimental outcomes, ES exhibited a moderating effect that was negative in the association between

COP and EMA adoption. According to the institutional theory literature, institutional pressures may lead to a positive or negative impact. If it is giving a positive impact, this means that the coercive rules and regulations are accepted by the community and have become a social norm in the organization [13,51,56]. While the Egyptian government pressures the industrial sector to comply with the newly issued environmental regulations, this hinders implementation as it faces implicit resistance, and companies try to tick boxes to tell the government that they are complying while they are not, which DiMaggio and Powell [36] call decoupling in the isomorphic processes. Contrary to the positive impact that institutional pressures have on the change process, this result conforms to many studies in the literature that discuss decoupling in practice as a response to institutional pressures [13,22,51,56,66,67]. The decoupling emanates from companies in developing countries trying to show the stakeholders that they are implementing, when they are not, to gain legitimacy. The actual practices that are occurring underneath appear when there is a clear ES that the management wants to apply to react to COP. Here, it is apparent that these strategies make the implementation and adoption appear as box ticking and maneuvering techniques to gain legitimacy.

Regarding the moderating effect of ES in the relationship between MIP and EMA adoption, it is revealed that ES exhibited a moderating effect that was positive in the association between MIP and EMA adoption. According to the institutional theory, having a clear environmental strategy and finding a successful adopter of EMA will lead to better adoption by the mimicking companies and make their implementation and adoption much easier. This follows the early studies that explained that EMA adoption can be costly and financial returns uncertain [33,70], and therefore, learning from successful competitors becomes particularly valuable. MIP can thus drive firms to imitate the practices of successful EMA adopters, motivated by the potential to reduce exploration costs associated with researching alternative approaches and minimize experimentation costs associated with implementing untested solutions [7,33,35,55,60,70].

Finally, regarding the role of ES in the relationship between NOP and EMA adoption. the empirical results revealed that there was no moderating role for ES. Although previous literature has emphasized the importance of ES in improving the awareness and producing social norms [5,24,65,70], this study's findings diverge from some existing research, likely due to the recent emphasis on sustainability by the Egyptian government. The dominant pressure on Egyptian companies is currently coercive in nature. However, we anticipate a shift in EMA adoption dynamics over time. As successful examples emerge, social norms within the industrial sector will likely become increasingly influential, driving competition around EMA implementation. It is further expected that these constructed norms could eventually transform the moderating impact of ES on the relationship between COP and EMA adoption. Currently, ES appears to exhibit a negative impact on the COP–EMA adoption link; however, as norms solidify, we may observe a transition to a positive impact, with norms becoming the key change driver and reducing the need for direct governmental pressure.

## 7. Implications, Limitations, and Future Research

This study bears multiple theoretical implications. Its principal contribution lies in devising a conceptual model probing the relationships among IP's three pressures, ES, and EMA adoption within the Egyptian industrial sector. While previous studies have explored the association between IP and EMA adoption, this research distinguishes itself by centering on the strategic facets of these connections and investigating the extent to which EMA adoption could be changed and impacted by the existence of a clear ES.

Grounded in institutional theory, especially in the NIS version of the theory, this study reinforces prior findings indicating that directing organizational efforts towards bolstering robust ES augments sustainability through elevating better EMA adoption. Furthermore, this research illuminates the linkage between institutional pressures and EMA adoption and their interplay with ES, while early studies confirmed the direct impact of both ES

and EMA adoption on environmental performance and sustainable development, an area warranting future exploration in academia. Nonetheless, some findings deviated from the previous literature, particularly in revealing a lack of statistical significance for the interaction among ES, NOP, and EMA adoption.

This paper posits that institutional theory offers a more robust framework for understanding EMA adoption than explanations centered on technical rationality. The argument is that managers often implement EMA practices driven by social compliance and the internalization of norms, values, and assumptions, rather than purely for technical benefits. Our findings have important implications for practitioners and policymakers: to promote EMA adoption and improve environmental performance, strengthening institutions is key to providing the impetus for organizational change. Institutional pressures shape organizational norms, beliefs, and culture, fostering an environment that champions environmental protection and supports the integration of proactive environmental practices. Moreover, institutional pressure plays a crucial role in steering organizations towards the effective implementation of environmental management initiatives.

This study's findings highlight the potential synergy between environmental strategies (ESs) and improved corporate performance, suggesting that environmental initiatives can yield both environmental and economic benefits for firms. The research also emphasizes the crucial role of accounting information systems like EMA in supporting ES implementation. EMA provides valuable data for decision-making, planning, and control processes, aiming to achieve the combined goals of environmental protection and economic gains. These findings underscore the potential for enhancing corporate performance through a dual approach: (1) implementing well-defined EMA and (2) developing robust accounting systems aligned with environmental considerations. This necessitates resource allocation both towards ES implementation and the development of advanced sustainability-oriented accounting information systems. Additionally, continuous investment in employee training regarding ES, performance measures, and their practical applications is crucial. This combined approach can not only improve performance but also enable firms to achieve greater congruence with stakeholders' expectations related to sustainability.

This study's insights, while valuable, should be interpreted in light of several limitations. The reliance on cross-sectional data suggests caution when making broad generalizations. Longitudinal or panel data could yield a more nuanced understanding of the dynamic relationships between the constructs examined. Further, the study could benefit from exploring diverse contexts, countries, and cultures to gain a richer understanding of IP-EMA adoption relationships. Integrating qualitative methods (such as interviews) alongside existing quantitative approaches would add an important dimension to future investigations. Finally, future research should explore the potential mediating role of EMA adoption in the relationship between institutional pressures and corporate performance. This would deepen our understanding of causal links and provide a more comprehensive picture of the topic by incorporating insights and contextual information that quantitative data alone may not fully capture.

**Author Contributions:** Conceptualization, A.B.M.M. and M.A.; methodology, A.B.M.M. and M.A.; software, A.B.M.M. and M.A.; validation, A.B.M.M. and M.A.; analysis and interpretation of the data, A.B.M.M. and M.A.; the drafting of the paper, A.B.M.M. and M.A.; revising it critically for intellectual content, A.B.M.M. and M.A.; funding acquisition, A.B.M.M. and M.A. All authors have read and agreed to the published version of the manuscript.

**Funding:** This work was funded by the Deanship of Scientific Research, Vice Presidency for Graduate Studies and Scientific Research, King Faisal University, Saudi Arabia. [Project No. GRANT5950].

**Institutional Review Board Statement:** The study was conducted according to the guidelines of the Declaration of Helsinki and approved by the deanship of the scientific research ethical committee, King Faisal University (project number: 5950, date of approval: 1 March 2024).

**Informed Consent Statement:** Informed consent was obtained from all subjects involved in the study.

**Data Availability Statement:** Data are available upon request from researchers who meet the eligibility criteria. Kindly contact the corresponding author privately through e-mail.

**Conflicts of Interest:** The authors declare no conflicts of interest.

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
