# Peer review of "Institutional Pressures and Environmental Management Accounting Adoption: Do Environmental Strategy Matter?"

_sustainability, doi:10.3390/su16073020_

Round 1

Reviewer 1 Report

Comments and Suggestions for Authors

This paper reports on an interesting survey on the drivers of environmental management accounting adoption. EMA is a relevant tool for managerial decision making that aims to incorporate the environmental impact of a business. The focus is on the role of institutional pressures (coercive, mimetic and normative pressures) towards EMA, and on the moderating role of environmental strategy for this relationship. To capture the perspective of developing countries, the authors gather and analyze data from 491 Egyptian manufacturing companies. The results are very interesting: all three types of institutional pressures are relevant for EMA adoption. In terms of the role of environmental strategy, the results are nuanced: while the moderating role of environmental strategy on the impact of normative pressures is insignificant, the coercive pressures in this developing country lead to a decoupling, while environmental strategy reinforces the impact of mimetic pressures on EMA.

The study is novel because of its focus on developing countries, and the attention paid to environmental strategy as a potential moderator. While the extant literature provides contradictory results, the results of this study are nuanced and mixed.

A first suggestion I would like to make, is that there are a couple of interesting management accounting papers linking EMA and environmental strategy, like Cheng, Mandy M., Paolo Perego, and Naomi S. Soderstrom. "Sustainability and Management Accounting Research." Journal of Management Accounting Research 35.3 (2023): 1-11, and Perego, Paolo, and Frank Hartmann. "Aligning performance measurement systems with strategy: The case of environmental strategy." Abacus 45.4 (2009): 397-428. These articles are interesting references for situating the paper in the extant literature.

The theoretical foundations in the paper, using institutional theory, work well. Still, the authors can guide the reader a little more: the concepts coercive pressure, mimetic pressure, and normative pressure are not explicitly defined. It would be helpful to have them more clearly introduced and explained besides mentioning where they originate from.

The same goes for the notion of decoupling: it is first mentioned in the literature review (section 3.1). During my first reading of the article, I had not captured how important this notion would be, as it comes back in the discussion. I would recommend the authors to do a better job in explaining what decoupling exactly stands for.

A similar remark: in section 3.1.1.: the paper states: ‘although the theory suggests that COP primarily targets external aspects’ … : it is not clear to me what it means. The statement is very vague. It comes back on p.5, where I read that COP is not meant to directly target the companies, but that by impacting other actors, COP in the end has an indirect effect on companies. Is this correct? It would be nice to have this explained better.

I am very impressed by the high response rate, this is quite unusual in management accounting studies. It would be interesting to understand how the authors managed to get a response rate of 75,5%.

Is there a possibility to incorporate control variables in this type of study? I suspect that the size of the firm, its financial success, its industry, whether it has a foreign owner etc. can have a significant impact on EMA adoption.

Table 2 presents the different scales used in the survey. I find it unusual that some of the variables do do not seem to capture the variable itself, but the relationship with EMA adoption. For example: To measure COP, I agree to measure it as ‘our company must adhere to the stringent environmental regulations established by the local government’, and ‘companies that break environmental standard and regulations face a number of penalties’. But the first 2 capture the whole relationship between COP and the implementation of EMA. Is this methodologically correct?

A bit similar for the first item capturing NOP.

The discussion highlights the interesting positive moderating effect of environmental strategy on the relationship between mimetic pressures and EMA adoption. It brings back the three forms of mimetic behavior: frequency based, trait based, and outcome based. It is not clear why these three forms are highlighted again: am I correct that they did not appear in the analyses? Or do the authors have data about them? I’m sorry if I missed the point. 

Minor suggestions:

The paper is quite well written, but here and there the language deserves some attention.

-          The title is grammatically not correct: ‘does’ environmental strategy matter

-          P.5: line 208: the impact of this pressure might be ‘different’

-          P.14: line 511: organizations or companies

-          P.15: line 532: firms tend to follow and learn

-          Table 1: respondents’ profiles (not company profiles)

Comments on the Quality of English Language

see above: minor comments 

Author Response

Please find the attached file. We replied in table form and replied to each comment beside it.

Reviewer 2 Report

Comments and Suggestions for Authors

Thanks for working on a comparatively less developed economy to provide empirical evidence to demonstrate impacts from the different aspects of institutional theory to better explain the behaviour. It is a pleasure to read. 

Few minor things to note:

L29 Suggest 'manage' instead of 'tackle'

L34 suggest 'support' instead of 'backing'

Line 57 the abbreviation of LDC is already mentioned and used in Line 47

Line 80 Reference [5] need to be an in-text incorporated into the sentence

Line 73-76 Would be good to include a small paragraph to explain briefly (before the details in Section 2) why the paper is using institutional theory - eg. able to explore in more details from the three aspects which was discussed in Section 2 

Author Response

Please find the attached file. We replied in table form, and for each comment, we replied to it beside it.

Reviewer 3 Report

Comments and Suggestions for Authors

Author Response

Great Thanks to the editor and the three reviewers for their time and comments, which helped the researchers improve the quality of the present work until we reached this advanced stage with the research paper.

Moreover, many thanks for accepting the paper and seeing it from this positive perspective. Finally, the other two reviewers asked for minor changes in the paper, and we addressed the suggested revisions, made the required changes, and highlighted the changes in yellow.